# Comparison of analytical sensitivity of DNA-based and RNA-based nucleic acid amplification tests for reproductive tract infection pathogens: implications for clinical applications

Yu Ma,[1,2,3] Jian Jiang,[1,2,3] Yanxi Han,[1,3] Yuqing Chen,[1,2,3] Zhenli Diao,[1,2,3] Tao Huang,[1,2,3] Lei Feng,[1,2,3] Lu Chang,[1,2,3] Duo Wang,[1,2,3] Yuanfeng Zhang,[1,2,3] Jinming Li,[1,2,3] Rui Zhang[1,2,3]

**ABSTRACT**   Currently, DNA-based nucleic acid amplification tests (NAATs) and RNA-based NAATs are employed to detect reproductive tract infection (RTI) pathogens including *Chlamydia trachomatis* (*CT*), *Neisseria gonorrhoeae* (*NG*), and *Ureaplasma urealyticum* (*UU*). Although evaluations of DNA-based NAATs have already existed, the comparison of the two methods is scarce. Thus, we compared the limits of detection (LODs) of DNA-based and RNA-based NAATs on the same experimental conditions. Inactivated culture supernatants of *CT*, *NG,* and *UU* with determined pathogen DNA and RNA load were used to evaluate LODs of seven DNA kits and one RNA kit. The LODs of the seven DNA kits for *CT*, *NG,* and *UU* ranged between 38–1,480, 94–20,011, and 132–2,011 copies/mL, respectively. As for RNA kits, they could detect samples at RNA concentrations of 3,116, 2,509, and 2,896 copies/mL, respectively. The RNA concentrations of *CT*, *NG*, and *UU* were 40, 885, and 42 times that of corresponding pathogen DNA concentrations in the employed supernatants, so RNA kits could detect pathogen DNA concentrations as low as 78 copies/mL, 3 copies/mL, and 69 copies/mL, respectively, but the level of pathogen load that the RNA tests could detect was primarily dependent on the infectious phase and transcriptional level of RNA. Thus, a schematic of bacterial dynamics during the period of reproductive tract infections was provided, which suggests that in terms of the analytical sensitivity of pathogen detection, RNA tests are more suitable for detecting active infection and recovery phase, while DNA tests are more suitable for detection in the early stage of infection.

**IMPORTANCE**   Reproductive tract infections have considerable effects on the health of humans. *CT*, *NG* , and *UU* are common pathogens. Although evaluation of DNA-based tests has already existed, the comparison between DNA-based and RNA-based tests is rare. Therefore, this study compared the limits of detection of the two tests on the same experimental conditions. Results suggested that most DNA-based NAATs could detect *CT*, *NG*, and *UU* at DNA concentrations lower than 1,000 copies/mL, while RNA-based NAATs could detect bacteria at RNA concentrations around 3,000 copies/mL. Considering the copy number of RNA per bacterium is dynamic through the growth cycle, further comparison is combined with a schematic of bacterial dynamics. Results suggested that in terms of the analytical sensitivity of pathogen detection, RNA tests are more suitable for detecting active infection and recovery phase, while DNA tests are more suitable for detection in the early stage of infection.

**KEYWORDS**   reproductive tract infection pathogens, DNA-based NAAT, RNA-based NAAT, analytical sensitivity

Address correspondence to Jinming Li, jmli@nccl.org.cn, or Rui Zhang, ruizhang@nccl.org.cn.

Yu Ma and Jian Jiang contributed equally to this article. Author order was determined by the order of increasing seniority.

The authors declare no conflict of interest.

See the funding table on p. 11.

R eproductive tract infections (RTIs) and their complications have considerable effects on the health and well-being of men, women, adults, and newborns. Among RTIs, *Chlamydia trachomatis* (*CT*), *Neisseria gonorrhoeae* (*NG*), and *Ureaplasma urealyticum* (*UU*) are the common pathogens. As estimated by the World Health Organization (WHO), *CT* and *NG* resulted in 129 million and 82 million infections, respectively, in 2020 (1).

*CT* is an intracellular pathogen that can cause genital infection. Although the majority of infected patients show no clinical symptoms or abnormal physical exam findings (2, 3), untreated *CT* infection can lead to severe complications and multiple sequelae, causing permanent damage to the genital tract. Among men, urethritis accompanied by urethral discharge and dysuria is the typical symptom. *CT* infection during pregnancy might cause neonatal chlamydia conjunctivitis, pneumoniae, and even neonatal death (4). *NG* is a facultatively intracellular diplococcus that tends to grow and multiply in the mucous membranes of the reproductive tract such as the cervix, uterus, fallopian tubes, and urethra (5). Urethritis caused by *NG* is predominantly symptomatic with urethral pruritis, mucoid, and purulent discharge in men, while in women, most *NG* infections are asymptomatic and even when symptoms are present, they are commonly mild and nonspecific (6). However, if left untreated, *NG* can invade the genital tract and cause pelvic inflammatory disease (PID), which may lead to the same long-term sequelae as *CT* (7). Therefore, annual screening among sexually active and pregnant women under 25 yr and older women with risk factors is recommended by the Centers for Disease Control and Prevention (CDC) (6). *UU* is the smallest cell that can reproduce itself (8). Despite the fact that it is regarded as a symbiotic bacterium in the genital tract, it is reported to account for up to 30% nongonococcal urethritis (9, 10). Severe *UU* infections can also result in infertility, abortion, and neonatal infection (11). In order to screen effectively and confirm infected patients as soon as possible to provide timely appropriate clinical treatment and to prevent subsequent complications and sequelae, accurate and rapid detection methods are critical.

Traditionally, culture for *CT*, *NG*, and *UU* is the reference standard, but it is time-consuming, technically challenging, and insensitive (12). Subsequently developed nonculture diagnostic tests are convenient and easy to use, but numerous positive cases were missed (13–16). The introduction of nucleic acid amplification tests (NAATs) in 2002 revolutionized the detection of genital infection pathogens due to their high sensitivity, specificity, and short turn-around time, so they are recommended for the diagnosis of *CT* and *NG* infections by the CDC (17).

At present, DNA-based NAATs remain the mainstream detection method. So far, numerous DNA detection kits have been approved by the Food and Drug Administration (FDA). These DNA detection kits are designed to amplify stable and specific DNA sequences in the cryptic plasmid or chromosome (18, 19). Researches have confirmed that these DNA detection kits show superiority in overall performance when compared with other nonculture methods (20). However, recently an increasing number of researchers have transferred their interest to RNA-based NAATs due to the distinct characteristics of RNA. On one hand, ribosomal RNA (rRNA) is approximately 100–10,000 times the copy number of genomic DNA in bacteria, indicating that RNA-based NAATs may have higher sensitivity than DNA-based NAATs (21). On the other hand, studies have suggested that the expression of rRNA rather than DNA is strongly associated with bacterial metabolic activity, making it possible to discriminate established infections from residual nucleic acid in dead bacteria after successful treatment (22, 23).

In domestic, an RNA-based NAAT, simultaneous amplification testing method (SAT), relying on isothermal amplification of bacterial rRNA, has been developed by Shanghai Rendu Biotechnology Co., Ltd. Although several studies have claimed it to be more sensitive than DNA-based NAATs (24, 25), these evaluations are confined to comparing the detection rates of confirmed positive clinical samples, failing to explore the exact level of pathogen concentration that DNA-based NAATs and RNA-based NAATs can detect and further assess whether RNA-based NAATs can reach the detection limit of DNA-based NAATs. Accordingly, this study compared the limits of detection (LODs) of

DNA-based NAATs and RNA-based NAATs on the same experimental conditions using the same inactivated culture supernatants of *CT*, *NG*, and *UU* with determined DNA and RNA concentrations.

## MATERIALS AND METHODS

### Study design and samples

To evaluate the analytical sensitivity of DNA-based and RNA-based NAATs for *CT*, *NG*, and *UU*, inactivated culture supernatants of these three pathogens with determined concentrations were employed (Fig. 1). Inactivated culture supernatants of *CT* (ATCC VR-571B), *NG* (ATCC 19424), and *UU* (ATCC 27816) were provided by Shanghai Rendu Biotechnology Co., Ltd.

For comparable quantification among these different pathogens, plasmids containing the barcode sequence and target sequences were constructed by Sangon Biotech (Shanghai) Co., Ltd and linearized to calculate the amplification efficiency of quantitative PCR (qPCR) for each target using primers and probes designed by Oligo 7 software (DBA Oligo, Inc., USA) (Fig. 1A). The primers and probes used in quantification were targeted at *CT* 23S ribosomal DNA (rDNA), *NG* 16S rDNA, and *UU* 16S rDNA, respectively (Table S1; Table S2).

For DNA quantification, the pathogen DNA was extracted with the QIAamp DNA Mini Kit (Qiagen, Hilden, Germany). Subsequently, qPCR and droplet digital PCR (ddPCR) were performed on the Applied Biosystems 7500 Real-time PCR System (USA) and Bio-Rad QX-200 system (USA), respectively, using primers and probes verified above (Fig. 1B). For RNA quantification, the pathogen RNA was extracted with the Qiagen RNeasy RNA Kit (Qiagen, Hilden, Germany) and reverse transcribed into cDNA using the PrimeScript RT Reagent Kit (Perfect Real Time; TaKaRa, Japan). Afterward, qPCR and ddPCR were performed as described above (Fig. 1B). For the same pathogen, the quantitative targets, primers, and probes were the same for DNA quantification and RNA quantification (Table S2).

### Evaluation of the LODs of DNA-based NAATs

*CT*, *NG,* and *UU* culture supernatants were diluted at $6.561 \times 10^5$, $2.187 \times 10^5$, $7.29 \times 10^4$, $2.43 \times 10^4$, $8.1 \times 10^3$, $2.7 \times 10^3$, $9 \times 10^2$, and $3 \times 10^2$ copies/mL according to the quantitative results of rDNA, respectively. Subsequently, 200 µL supernatants at each concentration level were employed to extract DNA using the Tianlong automatic nucleic acid extraction system (NP968-C, Xi'an Tianlong Science and Technology Co., Ltd). Samples at each concentration level were tested 20 times using seven China's National Medical Products Administration (NMPA) approved DNA-based NAATs (Sansure Bio-tech Co., Ltd., Xi'an Tianlong Science and Technology Co., Ltd., Shanghai Liferiver BioTech Co., Ltd., Hangzhou ACON Biotech Co., Ltd., Jiangsu Bioperfectus Technologies Co., Ltd., Daan Gene Co., Ltd. of Sun Yat-sen University, and Fosun Diagnostics (Shanghai) Co., Ltd.) (Fig. 1C; Table 1). Among these DNA-based NAATs, kits from Xi'an Tianlong Science and Technology Co., Ltd., Shanghai Liferiver BioTech Co., Ltd., Hangzhou ACON Biotech Co., Ltd., and Daan Gene Co., Ltd. of Sun Yat-sen University claimed to centrifugally precipitate 1 mL sample to extract DNA, so compared with 200 µL sample, this step introduced a dilution of fivefold. This factor was taken into consideration when calculating the LODs afterward. Two negative control samples were applied for each batch of tests to monitor contamination in the process of detection. Retesting and interpretation of results were conducted following the manufacturers' instructions.

### Evaluation of the LOD of the RNA-based NAAT

*CT*, *NG,* and *UU* culture supernatants were diluted at $6.561 \times 10^5$, $2.187 \times 10^5$, $7.29 \times 10^4$, $2.43 \times 10^4$, $8.1 \times 10^3$, $2.7 \times 10^3$, $9 \times 10^2$, and $3 \times 10^2$ copies/mL according to the quantitative results of RNA, respectively. Subsequently, RNA was extracted using the Tianlong automatic nucleic acid extraction system (NP968-C, Xi'an Tianlong Science and

**TABLE 1** Characteristics of DNA-based and RNA-based NAATs for *CT, NG,* and *UU*[a]

| | Specimens | Claimed LOD | Template input | Total reaction vol | Total PCR cycle no. |
|---|---|---|---|---|---|
| | | DNA-based NAATs | | | |
| *CT* detection kits | | | | | |
| Sansure | urethral swab, cervical swab | 400 copies/mL | 5 µL | 50 µL | 45 |
| Tianlong | urethral swab, cervical swab | 500 copies/mL | 4 µL | 40 µL | 40 |
| Liferiver | urethra and genital tract secretions | 1,000 copies/mL | 4 µL | 40 µL | 40 |
| Acon | urethral swab, cervical swab | 500 copies/mL | 4 µL | 40 µL | 40 |
| Bioperfectus | urethral swab, cervical swab | 500 IFU/mL | 5 µL | 25 µL | 40 |
| Daan | genitourinary secretions | 10 CT/mL | 2 µL | 45 µL | 40 |
| Fosun | genitourinary tract swab | 100 bacteria/mL | 4 µL | 30 µL | 40 |
| *NG* detection kits | | | | | |
| Sansure | urethral swab, cervical swab | 400 copies/mL | 5 µL | 50 µL | 45 |
| Tianlong | urethral swab, cervical swab | 500 copies/mL | 4 µL | 40 µL | 40 |
| Liferiver | urethra and genital tract secretions | 1,000 copies/mL | 4 µL | 40 µL | 40 |
| Acon | urethral swab, cervical swab | 500 copies/mL | 4 µL | 40 µL | 40 |
| Bioperfectus | urethral swab, cervical swab | $1 \times 10^4$ bacteria/mL | 5 µL | 25 µL | 40 |
| Daan | genitourinary secretions | $1.1 \times 10^4$ copies/mL | 2 µL | 45 µL | 40 |
| Fosun | genitourinary tract swab | $1 \times 10^3$ bacteria/mL | 4 µL | 30 µL | 40 |
| *UU* detection kits | | | | | |
| Sansure | urethral swab, cervical swab | 400 copies/mL | 5 µL | 50 µL | 45 |
| Tianlong | urethral swab, cervical swab | 500 copies/mL | 4 µL | 40 µL | 40 |
| Liferiver | urethra and genital tract secretions | 1,000 copies/mL | 4 µL | 40 µL | 40 |
| Acon | urethral swab, cervical swab | 500 copies/mL | 4 µL | 40 µL | 40 |
| Bioperfectus | urethral swab, cervical swab | $1 \times 10^4$ CCU/mL | 5 µL | 25 µL | 40 |
| Daan | genitourinary secretions | $1 \times 10^4$ copies/mL | 2 µL | 45 µL | 40 |
| Fosun | genitourinary tract swab | 100 CCU/mL | 4 µL | 30 µL | 40 |
| | | RNA-based NAATs (Rendu) | | | |
| *CT* detection kit | urethral swab, cervical swab, urine | 1,000 copies/reaction | 30 µL | 40 µL | 40 |
| *NG* detection kit | urethral swab, cervical swab, urine | 1,000 copies/reaction | 30 µL | 40 µL | 40 |
| *UU* detection kit | urethral swab, cervical swab, urine | 50 CFU/reaction | 30 µL | 40 µL | 40 |

[a]NAAT, nucleic acid amplification test; CT, Chlamydia trachomatis; NG, Neisseria gonorrhoeae; UU, Ureaplasma urealyticum; and LOD, the limit of detection.

Technology Co., Ltd) with matched extraction reagents and tested 20 times for each concentration level using an NMPA-approved RNA-based NAAT (Shanghai Rendu Biotechnology Co., Ltd.) (Fig. 1C; Table 1). This RNA detection kit was developed and based on simultaneous amplification and testing, combining isothermal amplification of RNA and real-time detection of fluorescence. Two negative control samples were applied for each batch of tests to monitor contamination in the process of detection. Retesting and interpretation of results were conducted following the manufacturers' instructions.

## Statistical analysis

MedCalc Statistical Software version 20.0.26 (MedCalc Software Ltd, Ostend, Belgium) was used for probit regression analysis to estimate the LODs of seven DNA detection kits and one RNA detection kit.

## RESULTS

## Quantification of DNA and RNA for inactivated culture supernatants of *CT*, *NG,* and *UU*

According to standard curves established by plasmids using self-designed primers and probes, the correlation coefficient was greater than 0.99, the amplification efficiency of

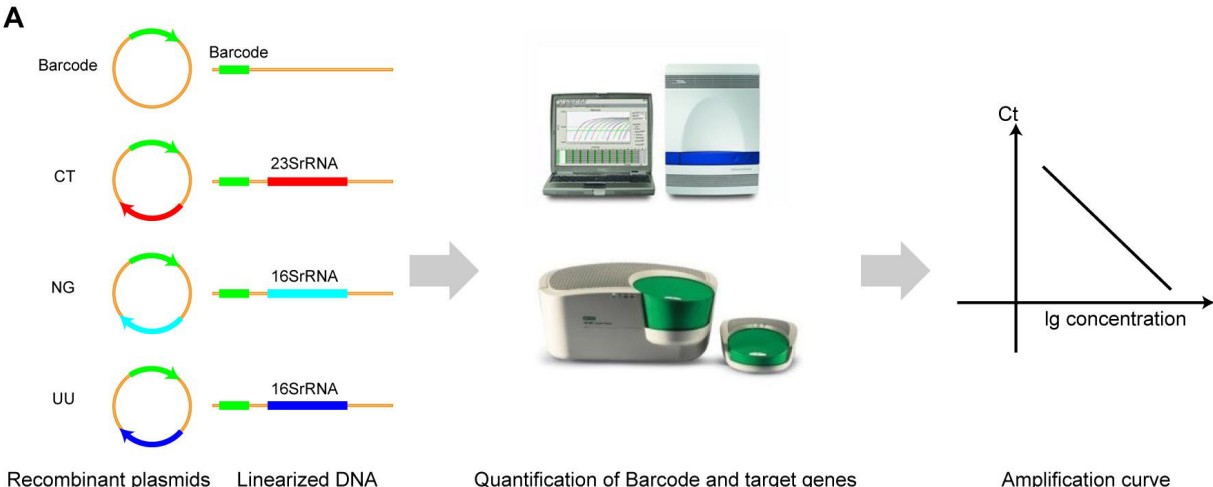

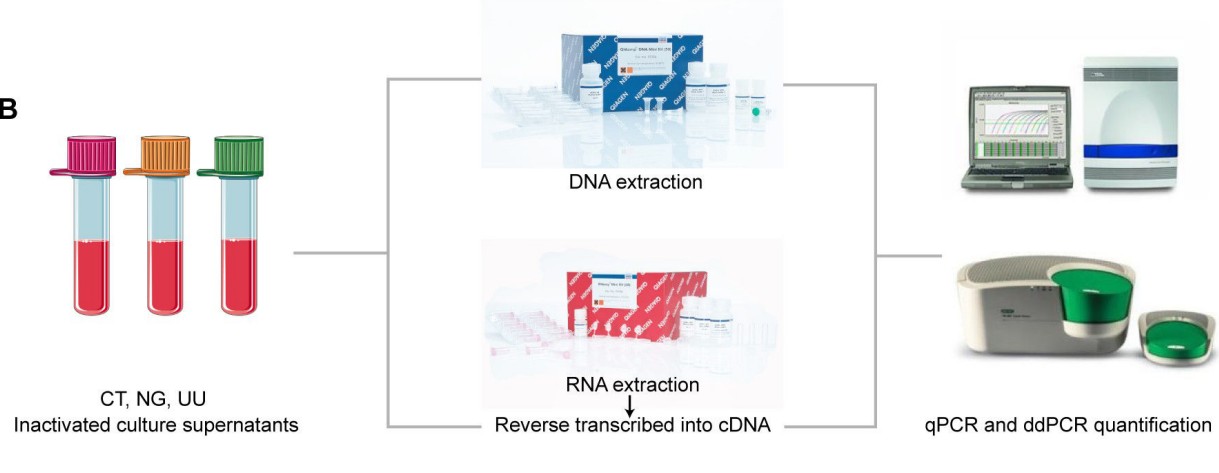

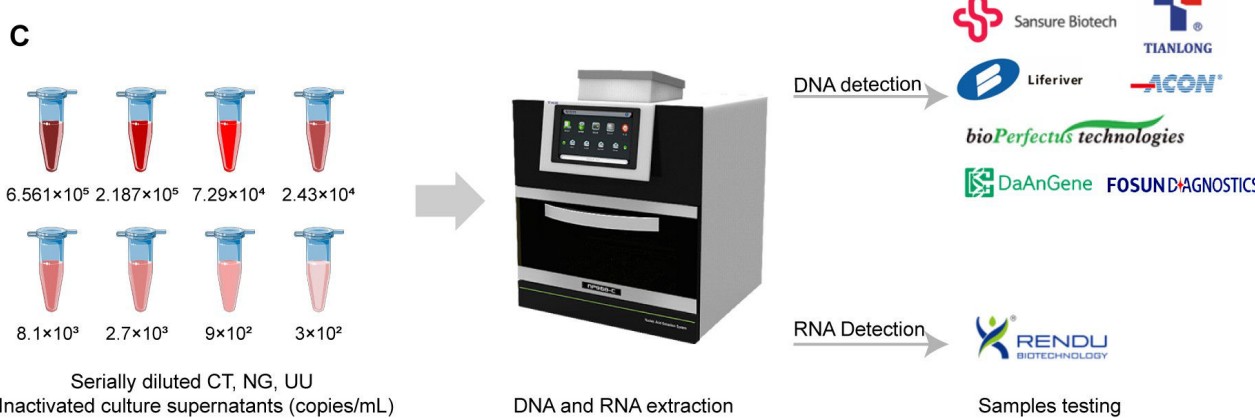

**FIG 1** Schematic diagram of the study design. (A) Calculation of amplification efficiency of quantitative PCR. (B) DNA and RNA quantification of *CT*, *NG*, and *UU* inactivated culture supernatants. (C) Evaluation of LODs of DNA-based and RNA-based NAATs.

qPCR was close to 100%, and the copy number ratio of quantitative target/barcode was close to 1 for each quantitative target (Table S3).

For inactivated culture supernatants of *CT*, the original concentrations of *CT* 23S rDNA and *CT* 23S rRNA were $4.232 \times 10^7$ and $8.42 \times 10^8$ copies/mL, respectively. Considering there are two copies of 23S rDNA in the *CT* genome, the pathogen load of *CT* was $2.116 \times 10^7$ copies/mL, indicating that the concentration of 23S rRNA was around 40 times that of pathogen DNA in the employed supernatants. For *NG*, the original 16S rDNA and 16S rRNA concentrations were $9.56 \times 10^8$ and $2.116 \times 10^{11}$ copies/mL, respectively. Considering there are four copies of 16S rDNA in the *NG* genome, the pathogen load of *NG* was $2.39 \times 10^8$ copies/mL, indicating that the concentration of 16S rRNA was around 885 times that of pathogen DNA in the employed supernatants. For *UU*, the original 16S rDNA and 16S rRNA concentrations were $5 \times 10^6$ and $1.05 \times 10^8$ copies/mL, respectively. Considering there are two copies of 16S rDNA gene in the *UU* genome, the pathogen load of *UU* was $2.5 \times 10^6$ copies/mL, indicating that the concentration of 16S rRNA was around 42 times that of pathogen DNA in the employed supernatants.

## Evaluation of the LODs of DNA-based NAATs

Probit regression analysis of the positive rate of 160 samples was conducted to calculate the LODs of DNA-based NAATs (Table 2). For *CT*, the LODs of Sansure, Tianlong, Liferiver, Acon, Bioperfects, Daan, and Fosun were 190, 38, 554, 295, 218, 216, and 1,480 copies/mL, respectively. For *NG*, the LODs of Sansure, Tianlong, Liferiver, Acon, Bioperfects, Daan, and Fosun were 111, 145, 94, 157, 11,985, 722, and 20,011 copies/mL, respectively. For *UU*, the LODs of Sansure, Tianlong, Liferiver, Acon, Bioperfects, Daan, and Fosun were 1,287, 132, 270, 244, 2,011, 287, and 650 copies/mL, respectively. The fivefold dilution effect of Tianlong, Liferiver, Acon, and Daan was considered when calculating the LODs. The results of the probit regression analysis are presented in Fig. 2.

## Evaluation of the LODs of the RNA-based NAATs

At the same time, probit regression analysis was conducted on the positive rate of 160 samples to calculate the LODs of the Rendu RNA Detection Kit (Table 3). The LODs of Rendu for *CT*, *NG*, and *UU* were 3,116, 2,509, and 2,896 copies/mL, respectively. The results of the probit regression analysis are presented in Fig. 2.

## DISCUSSION

RTIs remain a major and growing global public health concern, which considerably compromise female productivity, male fertility, and infant survival. Although most RTIs caused by *CT*, *NG*, and *UU* are curable with antibiotics, the untimely diagnosis and treatment due to the possibility of coinfection with other pathogens and the culture difficulty is likely to result in severe complications such as PID, infertility, and adverse pregnant outcomes (26). Therefore, early diagnosis is critical in decreasing infection time, reducing the spread of transmission, and minimizing sequelae. As recommended by the CDC, disease control strategies based on early diagnosis and treatment are significant determinants in relieving the burden of morbidity and mortality resulting from RTIs (27).

Aimed to detect *CT*, *NG*, and *UU* infection at the first time, numerous DNA-based NAATs have been developed. As of 24 June 2023, more than 20 DNA-based NAATs have been approved by the NMPA in China (28). Besides, due to the characteristic of rRNA as an indicator of bacterial metabolism, RNA-based NAATs for *CT*, *NG*, and *UU* detection have subsequently been designed by Shanghai Rendu Biotechnology Co., Ltd. So far, these NAATs have only been evaluated from the detection rate of positive clinical samples (29). Therefore, the performance of analytical sensitivity of both DNA-based NAATs and RNA-based NAATs remains to be assessed and compared. This study explored the analytical performance of seven DNA-based NAATs and one RNA-based NAAT approved by the NMPA. The *CT*, *NG*, and *UU* inactivated culture supernatants with determined pathogen loads by ddPCR were employed to acquire the analytical sensitivity.

**TABLE 2** Positive rates of seven DNA-based NAATs[a]

| | Number of positive/number of replicates (Positive rate, %) at diluted concentrations (copies/mL) | | | | | | | |
|---|---|---|---|---|---|---|---|---|
| Target DNA concentrations | $6.561 \times 10^5$ | $2.187 \times 10^5$ | $7.29 \times 10^4$ | $2.43 \times 10^4$ | $8.1 \times 10^3$ | $2.7 \times 10^3$ | $9 \times 10^2$ | $3 \times 10^2$ |
| CT concentrations | $3.2805 \times 10^5$ | $1.0935 \times 10^5$ | $3.645 \times 10^4$ | $1.215 \times 10^4$ | $4.05 \times 10^3$ | $1.35 \times 10^3$ | $4.5 \times 10^2$ | $1.5 \times 10^2$ |
| Sansure | 20/20 (100) | 20/20 (100) | 20/20 (100) | 20/20 (100) | 20/20 (100) | 20/20 (100) | 20/20 (100) | 13/20 (65) |
| Tianlong | 20/20 (100) | 20/20 (100) | 20/20 (100) | 20/20 (100) | 20/20 (100) | 20/20 (100) | 20/20 (100) | 13/20 (65) |
| Liferiver | 20/20 (100) | 20/20 (100) | 20/20 (100) | 20/20 (100) | 20/20 (100) | 16/20 (80) | 8/20 (40) | 3/20 (15) |
| Acon | 20/20 (100) | 20/20 (100) | 20/20 (100) | 20/20 (100) | 20/20 (100) | 20/20 (100) | 7/20 (35) | 3/20 (15) |
| Bioperfectus | 20/20 (100) | 20/20 (100) | 20/20 (100) | 20/20 (100) | 20/20 (100) | 20/20 (100) | 20/20 (100) | 5/20 (25) |
| Daan | 20/20 (100) | 20/20 (100) | 20/20 (100) | 20/20 (100) | 20/20 (100) | 20/20 (100) | 13/20 (65) | 6/20 (30) |
| Fosun | 20/20 (100) | 20/20 (100) | 20/20 (100) | 20/20 (100) | 20/20 (100) | 17/20 (85) | 0/20 (0) | 0/20 (0) |
| NG concentrations | $1.64025 \times 10^5$ | $5.4675 \times 10^4$ | $1.8225 \times 10^4$ | $6.075 \times 10^3$ | $2.025 \times 10^3$ | $6.75 \times 10^2$ | $2.25 \times 10^2$ | $7.5 \times 10^1$ |
| Sansure | 20/20 (100) | 20/20 (100) | 20/20 (100) | 20/20 (100) | 20/20 (100) | 20/20 (100) | 20/20 (100) | 4/20 (20) |
| Tianlong | 20/20 (100) | 20/20 (100) | 20/20 (100) | 20/20 (100) | 20/20 (100) | 20/20 (100) | 9/20 (45) | 5/20 (25) |
| Liferiver | 20/20 (100) | 20/20 (100) | 20/20 (100) | 20/20 (100) | 20/20 (100) | 20/20 (100) | 14/20 (70) | 5/20 (25) |
| Acon | 20/20 (100) | 20/20 (100) | 20/20 (100) | 20/20 (100) | 20/20 (100) | 20/20 (100) | 4/20 (20) | 2/20 (10) |
| Bioperfectus | 20/20 (100) | 20/20 (100) | 20/20 (100) | 20/20 (100) | 6/20 (30) | 4/20 (20) | 4/20 (20) | 0/20 (0) |
| Daan | 20/20 (100) | 20/20 (100) | 20/20 (100) | 20/20 (100) | 15/20 (75) | 7/20 (35) | 0/20 (0) | 0/20 (0) |
| Fosun | 20/20 (100) | 20/20 (100) | 20/20 (100) | 8/20 (40) | 5/20 (25) | 0/20 (0) | 0/20 (0) | 0/20 (0) |
| UU concentrations | $3.2805 \times 10^5$ | $1.0935 \times 10^5$ | $3.645 \times 10^5$ | $1.215 \times 10^4$ | $4.05 \times 10^3$ | $1.35 \times 10^3$ | $4.5 \times 10^2$ | $1.5 \times 10^2$ |
| Sansure | 20/20 (100) | 20/20 (100) | 20/20 (100) | 20/20 (100) | 20/20 (100) | 20/20 (100) | 8/20 (40) | 2/20 (10) |
| Tianlong | 20/20 (100) | 20/20 (100) | 20/20 (100) | 20/20 (100) | 20/20 (100) | 20/20 (100) | 4/20 (20) | 0/20 (0) |
| Liferiver | 20/20 (100) | 20/20 (100) | 20/20 (100) | 20/20 (100) | 20/20 (100) | 19/20 (100) | 3/20 (15) | 0/20 (0) |
| Acon | 20/20 (100) | 20/20 (100) | 20/20 (100) | 20/20 (100) | 20/20 (100) | 20/20 (100) | 7/20 (35) | 1/20 (5) |
| Bioperfectus | 20/20 (100) | 20/20 (100) | 20/20 (100) | 20/20 (100) | 20/20 (100) | 16/20 (80) | 2/20 (10) | 0/20 (0) |
| Daan | 20/20 (100) | 20/20 (100) | 20/20 (100) | 20/20 (100) | 20/20 (100) | 20/20 (100) | 6/20 (30) | 2/20 (10) |
| Fosun | 20/20 (100) | 20/20 (100) | 20/20 (100) | 20/20 (100) | 20/20 (100) | 20/20 (100) | 5/20 (25) | 0/20 (0) |

[a]NAAT, nucleic acid amplification test; CT, *Chlamydia trachomatis*; NG, *Neisseria gonorrhoeae*; and UU, *Ureaplasma urealyticum*.

For the quantification of *CT*, *NG,* and *UU* inactivated culture supernatants, good linearity and comparability were observed, with a correlation coefficient greater than 0.99, the amplification efficiency close to 100%, and a copy number ratio of quantitative target/barcode close to 1 for each quantitative target using corresponding plasmids. According to the quantification result, the rRNA concentrations of *CT*, *NG*, and *UU* were 40, 885, and 42 times that of corresponding pathogen DNA concentrations in the employed supernatants. The copy number of rRNA per bacterium is dynamic through the growth cycle, which is considered highly associated with the metabolic status. In this study, the employed inactivated culture supernatants contained 40–885 copies of rRNA per bacterium, basically consistent with relative researches that claimed the concentration of rRNA was approximately 100–10,000 times that of bacterial DNA (21, 24).

For detection of *CT*, *NG,* and *UU*, the LODs of the seven commonly used DNA detection kits on the Chinese market were found to range between 38–1,480, 94–20,011, and 132–2,011 copies/mL, respectively. Targeting different genes, the analytical sensitivity of these detection kits was not identical. Typically, targets for detection of *CT*, *NG,* and *UU* include specific chromosomal and cryptic plasmid sequences (17), and plasmid-based kits are proved to be more sensitive than chromosome-based kits due to the high copy number in bacteria (30). Generally, the bacterial load was more than $10^5$ copies/mL in swab samples of infected patients, which can be detected by these seven DNA detection kits (31–34). In terms of RNA detection kits, Rendu tests could detect samples at RNA concentrations of 3,116, 2,509, and 2,896 copies/mL for *CT*, *NG*, and *UU*, respectively. Due to the fact that the copy number of rRNA per bacterium is variable during the whole infectious progress, the level of pathogen load that the RNA tests could detect is primarily dependent on the infectious phase and transcriptional level of RNA. At the early stage of infection, the bacterium is at a low transcriptional level, and assuming that there

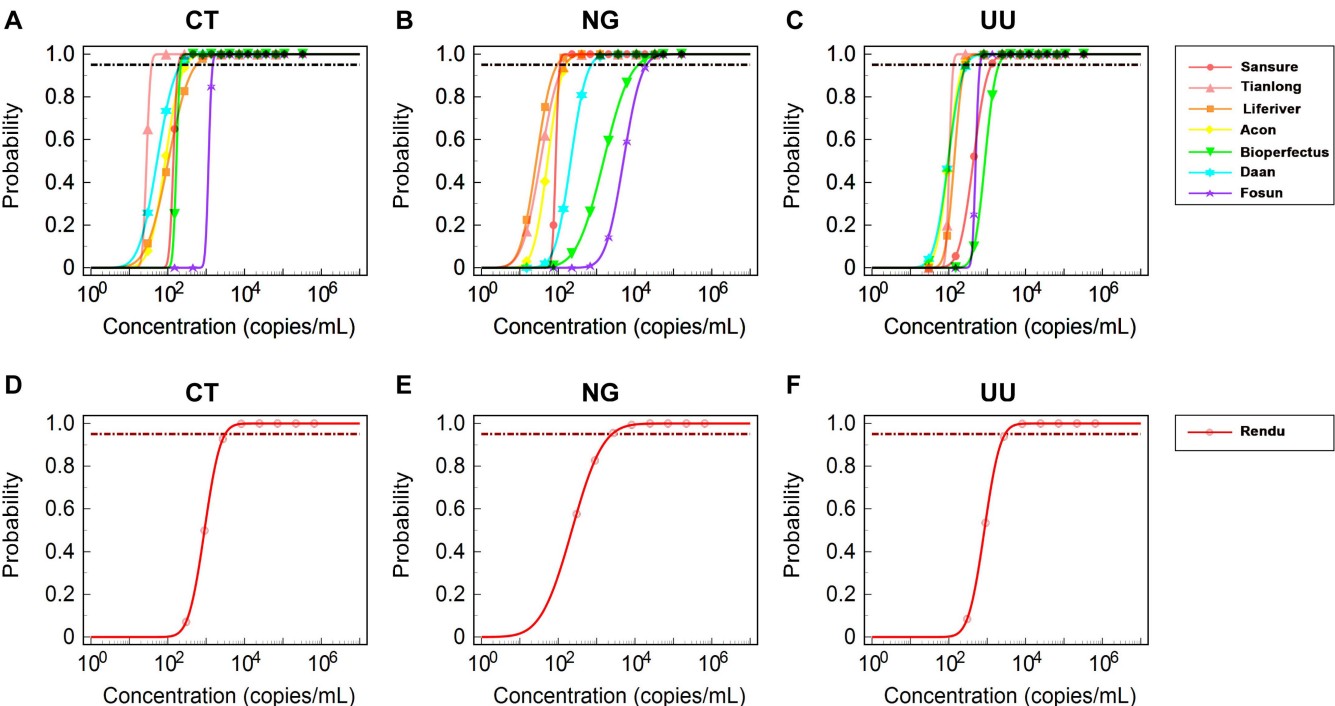

**FIG 2** Estimated limits of detection of the DNA-based and RNA-based NAATs using probit regression analysis. (A–C) Analysis graphs of probit regression of DNA-based NAATs for *CT*, *NG*, and *UU*, respectively. (D–F) Analysis graphs of probit regression of Rendu RNA-based NAAT for *CT*, *NG*, and *UU*, respectively.

is at least one rRNA transcript per bacterium, the highest DNA LODs could reach 3,116, 2,509, and 2,896 copies/mL for *CT*, *NG*, and *UU*, respectively. During active infection, when the copy number of rRNA could reach 10,000 per bacterium as is reported (21), the corresponding DNA LODs could be less than 1 copy/mL theoretically. When converting the RNA concentration level to pathogen DNA concentration level based on the copy number ratios of RNA/DNA of the employed supernatants, Rendu RNA Detection Kits could detect pathogen DNA concentrations as low as 78 copies/mL, 3 copies/mL, and 69 copies/mL for *CT*, *NG,* and *UU*, respectively (Fig. 3), but these results were confined to the employed supernatants.

Considering that our study is confined to inactivated culture supernatants, which couldn't represent the clinical urogenital samples during the whole infection, therefore, we collected data and information from peer researchers and drew a schematic of bacterial dynamics during the period of RTIs (Fig. 4). Studies show that at the early stage of infection, the incubation period is typically 5 to 10 d after exposure when a sharp increase of pathogen load can be observed. Subsequently, the infection reaches a stable phase over a period of 8 d, followed by a steady decline in pathogen load until resolution (32, 35, 36). In terms of rRNA expression, during the early stage of infection and recovery phase, the rRNA is at a low transcriptional level, while in active infection, the rRNA maintains a high transcriptional level and the copy number of rRNA is around 100–10,000fold greater than that of DNA (21, 22, 37–39). Combined with the LODs obtained in

**TABLE 3** Positive rates of Rendu RNA-based NAAT[a]

| Rendu RNA-based NAAT | Number of positive/number of replicates (Positive rate, %) at diluted concentrations (copies/mL) | | | | | | | |
|---|---|---|---|---|---|---|---|---|
| | RNA concentrations | | | | | | | |
| | $6.561 \times 10^5$ | $2.187 \times 10^5$ | $7.29 \times 10^4$ | $2.43 \times 10^4$ | $8.1 \times 10^3$ | $2.7 \times 10^3$ | $9 \times 10^2$ | $3 \times 10^2$ |
| *CT* detection kit | 20/20 (100) | 20/20 (100) | 20/20 (100) | 20/20 (100) | 20/20 (100) | 20/20 (100) | 6/20 (30) | 3/20 (15) |
| *NG* detection kit | 20/20 (100) | 20/20 (100) | 20/20 (100) | 20/20 (100) | 20/20 (100) | 19/20 (95) | 16/20 (80) | 12/20 (60) |
| *UU* detection kit | 20/20 (100) | 20/20 (100) | 20/20 (100) | 20/20 (100) | 20/20 (100) | 19/20 (95) | 10/20 (50) | 2/20 (10) |

[a]NAAT, nucleic acid amplification test; CT, *Chlamydia trachomatis*; NG, *Neisseria gonorrhoeae*; and UU, *Ureaplasma urealyticum*.

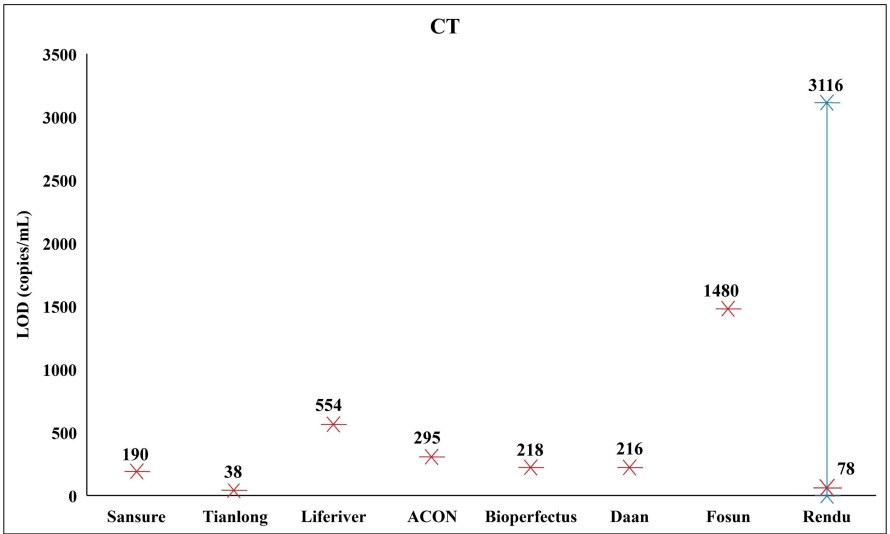

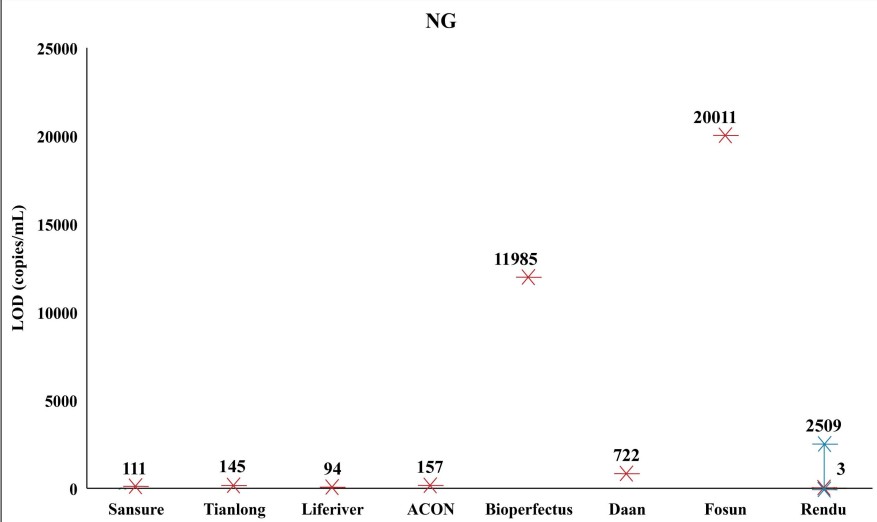

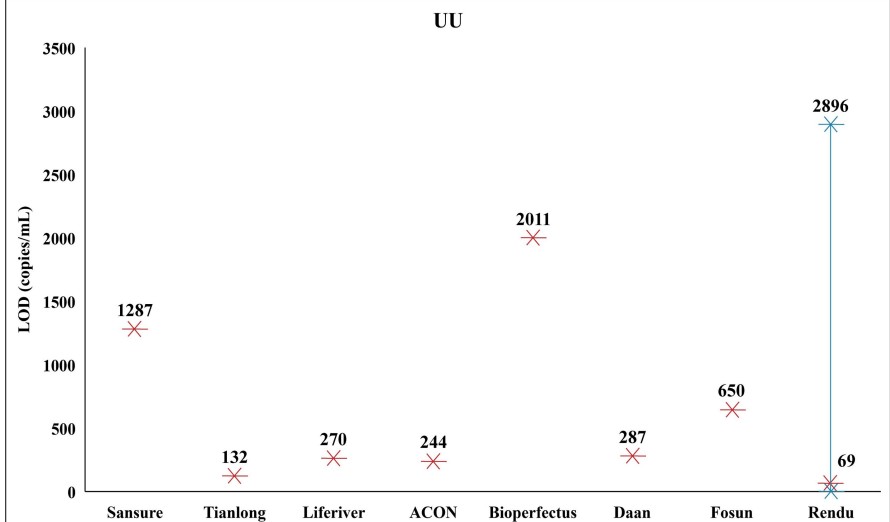

FIG 3 Limits of detection of the DNA-based and RNA-based NAATs for *CT*, *NG*, and *UU*, respectively. Red stars represent the LODs acquired based on the inactivated culture supernatants with determined DNA and RNA load. The span between blue stars represents the theoretical range of LODs of Rendu RNA Detection Kits dependent on the bacterial metabolic status.

our study, RNA tests could perform well during active infection, while at the early onset when the transcriptional level is low, Rendu RNA tests with LODs around 3,000 copies/mL may be inferior to some sensitive DNA tests with LODs lower than 100 copies/mL. However, in the recovery phase, rRNA, as a marker of bacterial metabolic status, can represent replication and transcriptional activity, thus truly reflecting the infection situation (22, 23). Therefore, in terms of the analytical sensitivity of pathogen detection, RNA-based NAATs are more suitable for detection during active infection and recovery phase, rather than in the early stage of infection. Compared with RNA-based NAATs, DNA-based NAATs are more suitable for detection in the early stage of infection.

At present, DNA-based NAATs remain the leading role in RTI detection both in domestic and abroad for their high sensitivity and specificity. However, most of them are targeted at the multicopy plasmids to improve sensitivity, which may lead to missed detections of plasmid-deficiency bacteria (30). Besides, on account of the residual nucleic acid of dead bacteria, positive results of *CT* and *NG* were still reported even after the cure, thus leading to the overtreatment and waste of medical resources (40). RNA-based NAATs, independent of plasmids, can complement the false negativity of many DNA-based NAATs in the presence of plasmid-deficiency pathogens. In terms of detection at the early onset, RNA-based NAATs are relatively sensitive when compared with some DNA-based NAATs with LODs close to 10,000 copies/mL. Additionally, closely associated with metabolic activity, RNA-based NAATs can discriminate active pathogens from dead ones, which is consistent with previous studies claiming that the clearance rate of rRNA was faster than DNA in patients during recovery (41). However, rRNA is subject to mutations more easily than DNA. In 2019, compromised detection of *CT* by the Aptima Combo 2 (AC2) diagnostic test (Hologic Inc., San Diego, California, United States [US]) targeting at *CT* 23S rRNA was reported due to a single base mutation (C1515T) in 23S rRNA, and this newly discovered variant strain of *CT* was announced as Finnish new

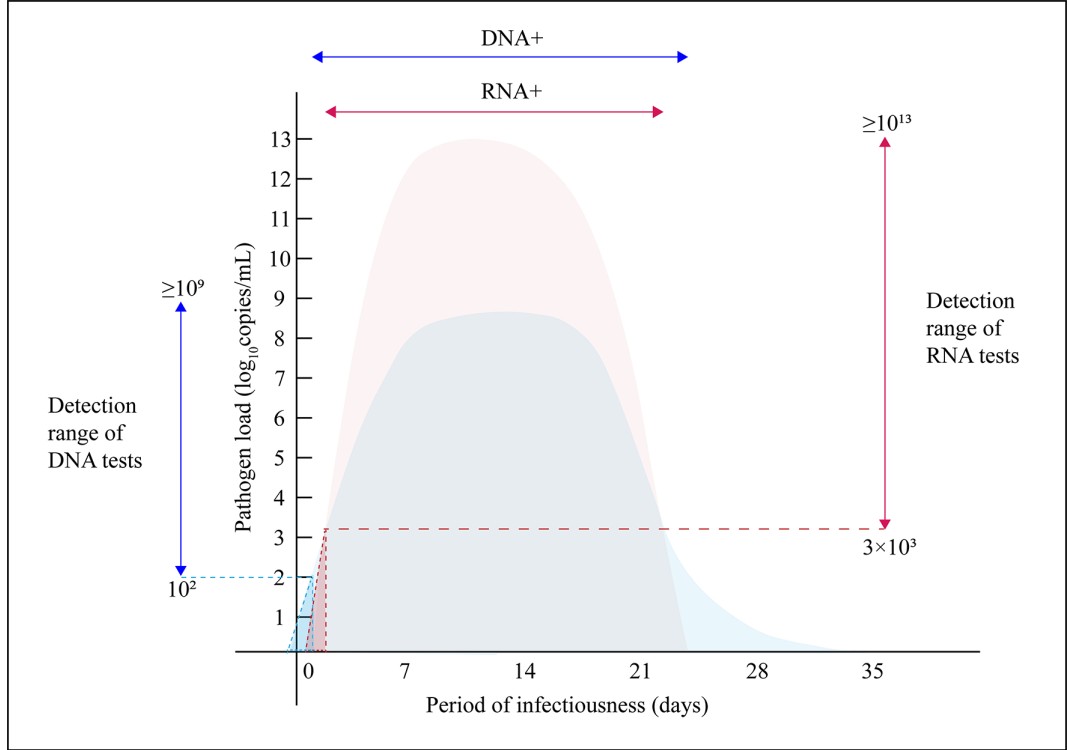

**FIG 4** Schematic of bacterial dynamics during the period of RTIs. The light blue area and light red area represent the dynamic variation of pathogen DNA and rRNA concentrations, respectively. The vertical blue line and red line with arrows represent the detection range of DNA and RNA tests, respectively. The horizontal blue and red lines with arrows represent the DNA-positive and RNA-positive periods, respectively. The blue triangular area depicted by the blue dashed lines and the red triangular area depicted by the red dashed lines represent the false negative area of DNA and RNA tests, respectively.

variant of *CT* (*FI-nvCT*) (42). Therefore, long-term surveillance of variants of reproductive tract pathogens is necessary and meaningful for RNA detection.

In conclusion, most DNA-based NAATs could detect *CT*, *NG*, and *UU* at DNA concentrations lower than 1,000 copies/mL, while RNA-based NAATs could detect bacteria at RNA concentrations of around 3,000 copies/mL. When converting the RNA concentration level to pathogen DNA concentration level using the copy number ratios of RNA/DNA based on the employed supernatants, RNA detection kits exhibited satisfactory analytical sensitivity with LODs as low as 78 copies/mL, 3 copies/mL, and 69 copies/mL for *CT*, *NG*, and *UU*, respectively, but the level of pathogen load that the RNA tests could detect was primarily dependent on the infectious phase and transcriptional level of RNA. As is described in the schematic of bacterial dynamics, during the early stage of infection, the rRNA is at a low transcriptional level where RNA-based tests show inferior performance in analytical sensitivity compared with DNA-based tests, while in active infection, the rRNA maintains a high transcriptional level so that RNA-based NAATs can detect pathogens at much lower concentrations compared to DNA-based NAATs (21, 22, 37–39). In the recovery phase, rRNA, as a marker of bacterial metabolic status, can represent replication and transcriptional activity, thus truly reflecting the infection situation (22, 23). Therefore, in terms of the analytical sensitivity of pathogen detection, RNA-based NAATs are more suitable for detection during active infection and recovery phase, rather than in the early stage of infection. Compared with RNA-based NAATs, DNA-based NAATs are more suitable for detection in the early stage of infection. As expatiated above, both DNA-based and RNA-based NAATs exist inherent drawbacks. Most DNA-based tests are targeted at plasmids, which may cause missed detection of plasmid-deficiency bacteria, and rRNA is subject to mutations, thus contributing to the missed detection of variants. Therefore, the long-term concern of plasmid-deficiency strains for DNA-based tests and surveillance of variants for RNA-based tests are indispensable and essential.

## ACKNOWLEDGMENTS

The authors would like to thank the Shanghai Rendu Biotechnology Co., Ltd. (China) for providing inactivated culture supernatants of *CT*, *NG,* and *UU* and thank all the reagent manufacturers that provided the automatic nucleic acid extraction system and NAATs for *CT*, *NG,* and *UU* detection in this study.

The authors declare no conflict of interest.

## AUTHOR AFFILIATIONS

[1]National Center for Clinical Laboratories, Institute of Geriatric Medicine, Chinese Academy of Medical Sciences, Beijing Hospital/National Center of Gerontology, Beijing, China

[2]National Center for Clinical Laboratories, Chinese Academy of Medical Sciences and Peking Union Medical College, Beijing, China

[3]Beijing Engineering Research Center of Laboratory Medicine, Beijing Hospital, Beijing, China

## AUTHOR ORCIDs

Yu Ma  http://orcid.org/0000-0003-4660-2042
Rui Zhang  http://orcid.org/0000-0003-4660-2042

## FUNDING

National Key R&D Program of China（2022YFC2603805）

## ADDITIONAL FILES

The following material is available online.

## Supplemental Material

**Supplemental information (Spectrum01497-23-S0001.pdf).** Tables S1 to S3.

## Open Peer Review

**PEER REVIEW HISTORY (review-history.pdf).** An accounting of the reviewer comments and feedback.

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
