## [Reviewer comments · Microbiology Spectrum]

Microbiology Spectrum

Comparison of analytical sensitivity of DNA-based and RNA-based nucleic acid amplification tests for reproductive tract infection pathogens: implications for clinical applications

Rui Zhang, Jinming Li, Yu Ma, Jian Jiang, Yanxi Han, Yuqing Chen, Zhenli Diao, Tao Huang, Lei Feng, Lu Chang, Duo Wang, and Yuanfeng Zhang

Corresponding Author(s): Rui Zhang, Graduate School, Peking Union Medical College, Chinese Academy of Medical Sciences, Beijing, People's Republic of China; National Center for Clinical Laboratories, Beijing Hospital

Review Timeline:

Submission Date:	April 9, 2023
Editorial Decision:	May 30, 2023
Revision Received:	June 24, 2023
Accepted:	July 1, 2023

Editor: Florence Doucet-Populaire

Reviewer(s): The reviewers have opted to remain anonymous.

Transaction Report:

DOI: <https://doi.org/10.1128/spectrum.01497-23>

May 30, 2023

Dr. Rui Zhang

Graduate School, Peking Union Medical College, Chinese Academy of Medical Sciences, Beijing, People's Republic of China;
National Center for Clinical Laboratories, Beijing Hospital
Beijing
China

Re: Spectrum01497-23 (Performance of DNA-based and RNA-based nucleic acid amplification tests for reproductive tract infection pathogens: implications for clinical applications)

Dear Dr. Rui Zhang:

Link Not Available

Sincerely,

Florence Doucet-Populaire

Journals Department
Reviewer comments:

Reviewer #1 (Comments for the Author):

The main limitation of this article is the absence of negative sample evaluations. Thus, a measure of the sensitivity and specificity of the tests is not possible, which would have been more interesting than a positivity rate. Furthermore, the authors repeatedly refer to the analytical sensitivity of the tests in the manuscript, which seems to me to be incorrect.

My other main point is that the authors chose to evaluate tests for *Ureaplasma urealyticum* (UU). This bacteria is responsible for urethritis in the first years of sexual life, but is subsequently considered a commensal germ. As such, it is rarely found in clinical practice. In my opinion, it would have been more interesting to evaluate tests for *Mycoplasma genitalium* (MG). RNA-based tests

in particular, because MG can be found in asymptomatic patients, and its treatment is very difficult because of frequent antibiotic resistance. So a test based on MG RNA could help assess the pathogenicity of MG and thus the need for treatment.

I also have a few minor comments :

Abstract:

- Page 2 line 33: the term RTI is not defined in the abstract.

Introduction:

- I find the introduction too long with a lot of information given by the authors (e.g. non-genital clinical manifestations of *Neisseria gonorrhoeae* (NG) or the sentence on antigenic detection tests) that are not essential to the understanding of the article.

- Page 3 line 48: Do not use the terms chlamydia and gonorrhoeae alone in this sentence. Instead, use the abbreviations you defined above.

- Page 3 line 52 to 57: I think the section on upper genital involvement of *Chlamydia trachomatis* (CT) in women should be reduced.

- Page 3 line 61: I disagree with this sentence. NG is predominantly symptomatic in men at the urethral site, with purulent discharge and mictional burning. In women, symptomatic cervicitis is frequent.

- Page 4, line 68: The authors state that UU is responsible for 70% of non-gonococcal urethritis, but I cannot find this information in the articles cited. Khosropour et al. gave a value of 22% in their study and Horner et al. gave a value of 30% in their table.

Discussion:

- Page 11 line 204: As stated above, NG is predominantly symptomatic in humans. This sentence should be rephrased.

- Page 12 line 228 to 241: This section should be included in the results section.

Reviewer #2 (Comments for the Author):

This article adds an important piece of information when considering RNA-based NAAT for the detection of urogenital pathogens. The authors make a solid point describing the relevance of their results, since, as far as they describe, no formal comparison between the analytic sensitivity of DNA and RNA NAAT has been previously published. Even more, in their discussion, they use their results to pinpoint important differences between both tests and explore the practical implications of their findings according to what is already known about the expected bacterial load during different stages of these infections.

I have some minor recommendations for the authors, mainly directed toward improving the presentation of their results:

1. My understanding is that the importance of this work is to compare the analytic sensitivity of an RNA-based test with the more widely available DNA-based NAAT tests for urogenital infections, in order to improve our understanding of the advantages and limitations of RNA NAAT in this setting. I think this is important since previous works measuring the analytic sensitivity of DNA NAAT already exist, and what the author's results add is the comparison between both tests on the same experimental conditions. I would suggest to the authors that this point is emphasized in the abstract and in the introduction (and possibly also in the title).

2. In the last paragraph, I would suggest the authors modify their conclusion, which currently states: "In conclusion, we evaluated the analytical sensitivity of seven DNA and one RNA detection kit for CT, NG, and UU by correlating pathogen replication levels with transcription levels through DNA and RNA quantification." The conclusion should not be just what was done, but what the implications of the results are. I think the question that needs to be explicitly answered by the authors and stated in the conclusion is: Do their results suggest that RNA NAAT and DNA NAAT have a comparable analytical sensitivity? If so, to all pathogens tested, or did they find any differences between them?

3. I also suggest the authors briefly highlight the main conclusion(s) of their results in the abstract and at the start of their discussion.

4. The "Importance" section of the abstract is almost exactly the same as the abstract, this needs to be modified according to the journal recommendations: "The Importance section should be no more than 150 words and should provide a nontechnical explanation of the significance of the study to the field. Avoid abbreviations and references and indicate the specific organism under study." (<https://journals.asm.org/journal/spectrum/article-types#research-articles>)

Minor formatting issues:

1. Bacterial species names should be in italics throughout the text.

2. In the abstract the RTI abbreviation is used before the use of the whole concept. This should be modified.

3. Line 235: missing word: "the NG load of was 2.39..."

4. Tables: All footnotes are missing the upper-case C in *Chlamydia trachomatis*

5. Table S2: The visual alignment between the Plasmid/Pathogen column and Primer/probe sequence column needs correction.

6. The text needs some minor English modifications as the use of "domestic", or the redaction of the last sentence of the paper: "As a matter of fact, both DNA and RNA-based NAATs exist inherent drawbacks such as missed detection...", which needs

clarification, among others.

Staff Comments:

Preparing Revision Guidelines

Please return the manuscript within 60 days; if you cannot complete the modification within this time period, please contact me. If you do not wish to modify the manuscript and prefer to submit it to another journal, please notify me of your decision immediately so that the manuscript may be formally withdrawn from consideration by Microbiology Spectrum.

This article adds an important piece of information when considering RNA-based NAAT for the detection of urogenital pathogens. The authors make a solid point describing the relevance of their results, since, as far as they describe, no formal comparison between the analytic sensitivity of DNA and RNA NAAT has been previously published. Even more, in their discussion, they use their results to pinpoint important differences between both tests and explore the practical implications of their findings according to what is already known about the expected bacterial load during different stages of these infections.

I have some minor recommendations for the authors, mainly directed toward improving the presentation of their results:

1. My understanding is that the importance of this work is to compare the analytic sensitivity of an RNA-based test with the more widely available DNA-based NAAT tests for urogenital infections, in order to improve our understanding of the advantages and limitations of RNA NAAT in this setting. I think this is important since previous works measuring the analytic sensitivity of DNA NAAT already exist, and what the author's results add is the comparison between both tests on the same experimental conditions. I would suggest to the authors that this point is emphasized in the abstract and in the introduction (and possibly also in the title).
2. In the last paragraph, I would suggest the authors modify their conclusion, which currently states: "In conclusion, we evaluated the analytical sensitivity of seven DNA and one RNA detection kit for CT, NG, and UU by correlating pathogen replication levels with transcription levels through DNA and RNA quantification." The conclusion should not be just what was done, but what the implications of the results are. I think the question that needs to be explicitly answered by the authors and stated in the conclusion is: Do their results suggest that RNA NAAT and DNA NAAT have a comparable analytical sensitivity? If so, to all pathogens tested, or did they find any differences between them?
3. I also suggest the authors briefly highlight the main conclusion(s) of their results in the abstract and at the start of their discussion.
4. The "Importance" section of the abstract is almost exactly the same as the abstract, this needs to be modified according to the journal recommendations: "The Importance section should be no more than 150 words and should provide a nontechnical explanation of the significance of the study to the field. Avoid abbreviations and references and indicate the specific organism under study." (<https://journals.asm.org/journal/spectrum/article-types#research-articles>)

Minor formatting issues:

1. Bacterial species names should be in italics throughout the text.

2. In the abstract the RTI abbreviation is used before the use of the whole concept. This should be modified.
3. Line 235: missing word: “the NG load of was 2.39...”
4. Tables: All footnotes are missing the upper-case C in Chlamydia trachomatis
5. Table S2: The visual alignment between the Plasmid/Pathogen column and Primer/probe sequence column needs correction.
6. The text needs some minor English modifications as the use of “domestic”, or the redaction of the last sentence of the paper: “As a matter of fact, both DNA and RNA-based NAATs exist inherent drawbacks such as missed detection...”, which needs clarification, among others.

Response to Reviewers

Dear Editors,

Thank you very much for your email dated May 30, 2023. We have revised the manuscript according to the comments of the reviewers, and the amendments are highlighted in RED in the “Marked Up Manuscript - For Review Only”. We also responded to the comments as listed below. The comments are all valuable and very helpful for improving our paper. We would like to thank the reviewers for the constructive and instructive comments.

With best wishes,

Yours sincerely,

Rui Zhang

Replies to the reviewer's comments:

Reviewer #1:

Comment 1: The main limitation of this article is the absence of negative sample evaluations. Thus, a measure of the sensitivity and specificity of the tests is not possible, which would have been more interesting than a positivity rate. Furthermore, the authors repeatedly refer to the analytical sensitivity of the tests in the manuscript, which seems to me to be incorrect.

Answer 1: Thank you for your valuable suggestion.

This study was primarily focused on the limit of detection, also known as the analytical sensitivity of the tests. Inactivated culture supernatants were diluted at 6.561×10^5 , 2.187×10^5 , 7.29×10^4 , 2.43×10^4 , 8.1×10^3 , 2.7×10^3 , 9×10^2 , 3×10^2 copies/mL to acquire the limits of detection of tests. Two negative samples composed of deionized water were tested in each batch of detection, but negative samples such as interfering and cross-reacting substances were not included. Therefore, the analytical specificity was not evaluated. Considering this factor, we have revised our title.

The modified title is as follows:

“Comparison of analytical sensitivity of DNA-based and RNA-based nucleic acid amplification tests for reproductive tract infection pathogens: implications for clinical applications.” (Page 1 line 1-3)

Comment 2: My other main point is that the authors chose to evaluate tests for *Ureaplasma urealyticum* (UU). This bacteria is responsible for urethritis in the first years of sexual life, but is subsequently considered a commensal germ. As such, it is rarely found in clinical practice. In my opinion, it would have been more interesting to evaluate tests for *Mycoplasma genitalium* (MG). RNA-based tests in particular, because MG can be found in asymptomatic patients, and its treatment is very difficult

because of frequent antibiotic resistance. So a test based on MG RNA could help assess the pathogenicity of MG and thus the need for treatment.

Answer 2: Thank you for your valuable suggestion.

Honestly, *Ureaplasma urealyticum* (*UU*) is a commensal organism of the genital tract, and researches have indicated that *UU* only causes urethritis when present in a high load, but studies have demonstrated that *UU* could be detected in the lower genital tract of 40–80% of sexually active women, and it might cause adverse pregnant outcomes such as infertility, preterm labor, and delivery [1]. Besides, its colonization in the cervix during pregnancy is associated with a high morbidity. Therefore, screening tests in pregnant women and patients with infertility have been proposed by several investigators [2-4]. Regarding *Mycoplasma genitalium* (*MG*), it could be detected in 1% to 3.3% of the general population and asymptomatic infections are frequent[5]. If symptoms are present, they are commonly accompanied by non-gonococcal urethritis in men, cervicitis, and pelvic inflammatory disease (PID) in women. Guidelines suggest that testing and treatment of asymptomatic *MG* infections are not recommended due to insufficient clinical research evidence [5]. In clinical practice, it should be suspected in cases of persistent or recurrent urethritis or cervicitis and considered for pelvic inflammatory disease (PID). Moreover, studies have indicated that *MG* has little association with adverse pregnancy and infertility, so screening tests are unnecessary among pregnant women and infertile patients [6]. In terms of the development of detection kits, so far, only two DNA-based *MG* tests have been approved by the China National Medical Products Administration (NMPA), while for DNA-based *UU* tests, more than twenty kits have been approved and are available. Considering the factors mentioned above, we have ultimately chosen to evaluate the nucleic acid amplification tests (NAATs) of *UU* instead of *MG*.

Comment 3: Page 2 line 33: the term RTI is not defined in the abstract.

Answer 3: Thanks for pointing this out.

We are sorry for this mistake. According to your suggestion, we have added the definition of the term RTI in the abstract.

The modified content is as follows:

“Thus, a schematic of bacterial dynamics during the period of **reproductive tract infections (RTIs) was provided, which suggesting that in terms of the analytical sensitivity of pathogen detection, RNA tests are more suitable for detecting active infection and recovery phase, while DNA tests are more suitable for detection in the early stage of infection.”** (Abstract, Page 2 line 37-41)

Comment 4: I find the introduction too long with a lot of information given by the authors (e.g. non-genital clinical manifestations of *Neisseria gonorrhoeae* (NG) or the sentence on antigenic detection tests) that are not essential to the understanding of the article.

Answer 4: Thank you for your valuable suggestion.

We have refined the introduction and removed the redundant information according to your suggestion. First, the content related to the non-genital clinical manifestations of *Neisseria gonorrhoeae* (Among men, the clinical manifestation comes with occasionally epididymitis complicated by arthritis, meningitis or endocarditis when *NG* spread to the blood and lead to disseminated gonococcal infection (DGI).) was removed and relative clinical manifestations were corrected according to suggestions in **Comment 7**. Then the description of antigen detection tests was deleted (Thus, nonculture diagnostic tests such as antigen detection tests and nucleic acid hybridization tests have been developed to provide convenient and easy-to-use alternatives. However, the primary drawback of these tests lies in their failure to test a large number of infections.) and the development of detection methods was condensed and shortened.

The modified content is as follows:

“*NG* is a facultatively intracellular diplococcus that tends to grow and multiply in the mucous membranes of the reproductive tract such as the cervix, uterus,

fallopian tubes, and urethra. Urethritis caused by *NG* is predominantly symptomatic with urethral pruritis, mucoid and purulent discharge in men, while in women, most *NG* infections are asymptomatic and even when a woman has symptoms, they are commonly mild and nonspecific. But if left untreated, *NG* can invade the genital tract and cause pelvic inflammatory disease (PID), which may lead to the same long-term sequelae as *CT*. Therefore, annual screening among sexually active and pregnant women under 25 years and older women with risk factors is recommended by the Centers for Disease Control and Prevention (CDC)." (Introduction, Page 4 line 69-79)

"Traditionally, culture for *CT*, *NG*, and *UU* is the reference standard, but it is time-consuming, technically challenging and insensitive. Subsequently developed nonculture diagnostic tests are convenient and easy-to-use, but numerous positive cases were missed. The introduction of nucleic acid amplification tests (NAATs) in 2002 revolutionized the detection of genital infection pathogens due to their high sensitivity, specificity and short turn-around time, so they are recommended for the diagnosis of *CT* and *NG* infections by the CDC." (Introduction, Page 5 line 86-92)

Comment 5: Page 3 line 48: Do not use the terms chlamydia and gonorrhoeae alone in this sentence. Instead, use the abbreviations you defined above.

Answer 5: Thanks for pointing this out.

We are sorry for these mistakes. According to your suggestion, these terms have been replaced with the abbreviations in the introduction.

The modified content is as follows:

"As is estimated by the World Health Organization (WHO), *CT* and *NG* resulted in 129 million and 82 million infections respectively in 2020." (Introduction, Page 4 line 60-62)

Comment 6: Page 3 line 52 to 57: I think the section on upper genital involvement of *Chlamydia trachomatis* (CT) in women should be reduced.

Answer 6: Thank you for your valuable suggestion.

According to your suggestion, the introduction is lengthy and some information is inessential to the understanding of the article, so we have removed the sentence related to the upper genital involvement of *Chlamydia trachomatis* (In previous observational studies, approximately 30% of women developed pelvic inflammatory disease (PID) when left untreated and some might even subsequently experience infertility, ectopic pregnancy, and chronic pelvic pain.) in the introduction.

The modified content is as follows:

“CT is an intracellular pathogen that can cause genital infection. Although the majority of infected patients show no clinical symptom or abnormal physical exam findings, untreated CT infection can lead to severe complications and multiple sequelae, causing permanent damage to the genital tract. Among men, urethritis accompanied by urethral discharge and dysuria is the typical symptom. CT infection during pregnancy might cause neonatal chlamydia conjunctivitis, pneumoniae, and even neonatal death.” (Introduction, Page 4 line 63-69)

Comment 7: Page 3 line 61: I disagree with this sentence. NG is predominantly symptomatic in men at the urethral site, with purulent discharge and mictional burning. In women, symptomatic cervicitis is frequent.

Answer 7: Thank you for your valuable suggestion.

As you pointed out, the description of symptoms of *NG* (Similar to *CT*, most patients infected with *NG* are asymptomatic.) is improper. To figure out the specific symptoms of *NG* infection, we referred to the *Sexually Transmitted Infections Treatment Guideline, 2021* issued by the Centers for Disease Control and Prevention (CDC) and learned that urethral infections caused by *NG* are predominantly symptomatic in men,

so they could seek treatment timely. While in women, most *NG* infections are asymptomatic and even when symptoms are present, they are commonly mild and nonspecific. Therefore, annual screening among sexually active and pregnant women under 25 years and older women with risk factors is recommended by the CDC. Accordingly, we have revised the corresponding content.

The modified content is as follows:

“Urethritis caused by *NG* is predominantly symptomatic with urethral pruritis, mucoid and purulent discharge in men, while in women, most *NG* infections are asymptomatic and even when symptoms are present, they are commonly mild and nonspecific [6]. But if left untreated, *NG* can invade the genital tract and cause pelvic inflammatory disease (PID), which may lead to the same long-term sequelae as *CT* [7]. Therefore, annual screening among sexually active and pregnant women under 25 years and older women with risk factors is recommended by the Centers for Disease Control and Prevention (CDC) [6].”

(Introduction, Page 4 line 71-79).

Reference:

- 6. Workowski, K.A., et al., *Sexually transmitted infections treatment guidelines, 2021. MMWR Recommendations and Reports, 2021. 70(4): p. 1.***
- 7. Ramezani, M., et al., Survey on the prevalence of Chlamydia trachomatis and Neisseria gonorrhoeae infections and their possible effects on seminal quality in infertile men. *International Journal of Infection, 2019. 6(4).***

Comment 8: Page 4, line 68: The authors state that UU is responsible for 70% of non-gonococcal urethritis, but I cannot find this information in the articles cited. Khosropour et al. gave a value of 22% in their study and Horner et al. gave a value of 30% in their table.

Answer 8: Thanks for pointing this out.

We are sorry for this mistake. Instead of the results of an article, the clarification that UU could account for 70% of non-gonococcal urethritis was a description mentioned

in the introduction of the article by citing others' references, and we indirectly cited these references it listed without checking. After verification of the references, we found the ratio should be up to 30%, so we corrected it in the manuscript.

The modified content is as follows:

“Despite the fact that it is regarded as a symbiotic bacterium in the genital tract, it is reported to account for up to 30% nongonococcal urethritis.” (Introduction, Page 4-5 line 79-81)

The article that claimed the ratio to be 70% is listed here:

Reference: Liu, T., et al., *Analysis of Ureaplasma urealyticum, Chlamydia trachomatis, Mycoplasma genitalium and Neisseria gonorrhoeae infections among obstetrics and gynecological outpatients in southwest China: a retrospective study.* BMC Infectious Diseases, 2022. **22**(1): p. 283.

Comment 9: Page 11 line 204: As stated above, NG is predominantly symptomatic in humans. This sentence should be rephrased.

Answer 9: Thanks for pointing this out.

As is discussed in the **Comment 7**, the clarification of NG infection as asymptomatic is improper, so we have deleted the expression “clinically silent nature” in the discussion (Although most RTIs caused by CT, NG and UU are curable with antibiotics, the untimely diagnosis on account of their clinically silent nature, the possibility of coinfection with other pathogens, and the culture difficulty is likely to result in severe complications.). Instead, we have focused on describing the severe complications that can arise from the untimely diagnosis and treatment of reproductive tract infections, considering factors such as the possibility of coinfection with other pathogens and the difficulty in culturing these organisms.

The modified content is as follows:

“Although most RTIs caused by CT, NG and UU are curable with antibiotics, the untimely diagnosis and treatment due to the possibility of coinfection with other pathogens, and the culture difficulty is likely to result in severe

complications such as PID, infertility, and adverse pregnant outcomes.”
(Discussion, Page 12 line 220-224).

Comment 10: Page 12 line 228 to 241: This section should be included in the results section.

Answer 10: Thank you for your valuable suggestion.

1. This section demonstrates the copy number ratio of RNA/DNA based on the quantification result of the inactivated culture supernatants, so it would be more appropriate to be present in the results section as you suggested.

The modified content is as follows:

“For inactivated culture supernatants of *CT*, the original concentrations of *CT* 23S rDNA and *CT* 23S rRNA were 4.232×10^7 and 8.42×10^8 copies/mL, respectively. Considering there are two copies of 23S rDNA in the *CT* genome, the pathogen load of *CT* was 2.116×10^7 copies/mL, indicating that the concentration of 23S rRNA was around 40 times that of pathogen DNA in the employed supernatants. For *NG*, the original 16S rDNA and 16S rRNA concentrations were 9.56×10^8 and 2.116×10^{11} copies/mL, respectively. Considering there are four copies of 16S rDNA in the *NG* genome, the pathogen load of *NG* was 2.39×10^8 copies/mL, indicating that the concentration of 16S rRNA was around 885 times that of pathogen DNA in the employed supernatants. For *UU*, the original 16S rDNA and 16S rRNA concentrations were 5×10^6 and 1.05×10^8 copies/mL, respectively. Considering there are two copies of 16S rDNA gene in the *UU* genome, the *UU* load was 2.5×10^6 copies/mL, indicating that the concentration of 16S rRNA was around 42 times that of pathogen DNA in the employed supernatants.” (Results, Page 10 line 188-201)

2. Additionally, to maintain the structural integrity of the discussion after the deletion of the section, the quantification result was further summarized and analyzed in combination with relevant researches.

The modified content is as follows:

“For quantification of *CT*, *NG* and *UU* inactivated culture supernatants, good linearity and comparability were shown with the correlation coefficient greater than 0.99, the amplification efficiency close to 100%, and the copy number ratio of quantitative target/barcode close to 1 for each quantitative target using corresponding plasmids. According to the quantification result, the rRNA concentrations of *CT*, *NG*, and *UU* were 40, 885, and 42 times that of corresponding pathogen DNA concentrations in the employed supernatants. The copy number of rRNA per bacterium is dynamic through the growth cycle, which is considered highly associated with the metabolic status. In this study, the employed inactivated culture supernatants contained 40-885 copies of rRNA per bacterium, basically consistent with relative researches which claimed that the concentration of rRNA was approximately 100-10000 times that of bacterial DNA.” (Discussion, Page 12-13 line 241-252)

Reviewer #2:

Comment 1: My understanding is that the importance of this work is to compare the analytic sensitivity of an RNA-based test with the more widely available DNA-based NAAT tests for urogenital infections, in order to improve our understanding of the advantages and limitations of RNA NAAT in this setting. I think this is important since previous works measuring the analytic sensitivity of DNA NAAT already exist, and what the author's results add is the comparison between both tests on the same experimental conditions. I would suggest to the authors that this point is emphasized in the abstract and in the introduction (and possibly also in the title).

Answer 1: Thanks for your endorsement and we really appreciate your positive and constructive comments and suggestions, which help us to improve our present manuscript. At present, evaluations of DNA-based NAATs for reproductive tract infections already exist, but the comparison of DNA and RNA-based tests is rare.

Thus, this study was focused on the comparison of the analytical sensitivity of widely acceptable DNA-based tests and an RNA-based test to provide possible implications for clinical applications. As you pointed out, the significance of this study wasn't fully demonstrated and highlighted, so we have made emphasis in the abstract, introduction, and title.

The modified content is as follows:

1. Abstract:

“Although evaluations of DNA-based NAATs have already existed, the comparison of the two methods is scarce. Thus, we compared the limits of detection (LODs) of DNA-based and RNA-based NAATs on the same experimental conditions.” (Abstract, Page 2 line 24-27)

2. Introduction

“Although several studies have claimed it to be more sensitive than DNA-based NAATs, these evaluations are confined to comparing the detection rates of confirmed positive clinical samples, failing to explore the exact level of pathogen concentration that DNA-based NAATs and RNA-based NAATs can detect and further assess whether RNA-based NAATs can reach the detection limit of DNA-based NAATs. Accordingly, this study compared the limits of detection (LODs) of DNA-based NAATs and RNA-based NAATs on the same experimental conditions using the same inactivated culture supernatants of *CT*, *NG*, and *UU* with determined DNA and RNA concentrations.” (Introduction, Page 6 line 109-117)

3. Title

“Comparison of analytical sensitivity of DNA-based and RNA-based nucleic acid amplification tests for reproductive tract infection pathogens: implications for clinical applications.” (Page 1 line 1-3)

Comment 2: In the last paragraph, I would suggest the authors modify their conclusion, which currently states: "In conclusion, we evaluated the analytical sensitivity of seven DNA and one RNA detection kit for CT, NG, and UU by correlating pathogen replication levels with transcription levels through DNA and

RNA quantification." The conclusion should not be just what was done, but what the implications of the results are. I think the question that needs to be explicitly answered by the authors and stated in the conclusion is: Do their results suggest that RNA NAAT and DNA NAAT have a comparable analytical sensitivity? If so, to all pathogens tested, or did they find any differences between them?

Answer 2: Thank you for your valuable suggestion.

As you pointed out, the last paragraph just briefly described the work we've completed, and the implications behind the results were not fully stated. Therefore, we have further demonstrated the analytical sensitivity of DNA and RNA-based NAATs based on the results and bacterial dynamics during the period of infection in the last paragraph.

The modified content is as follows:

"In conclusion, most DNA-based NAATs could detect *CT*, *NG*, and *UU* at DNA concentrations lower than 1000 copies/mL, while RNA-based NAATs could detect bacteria at RNA concentrations around 3000 copies/mL. When converting the RNA concentration level to pathogen DNA concentration level using the copy number ratios of RNA/DNA based on the employed supernatants, RNA detection kits exhibited satisfactory analytical sensitivity with LODs as low as 78 copies/mL, 3 copies/mL, and 69 copies/mL for *CT*, *NG* and *UU*, respectively, but the level of pathogen load that the RNA tests could detect was primarily dependent on the infectious phase and transcriptional level of RNA. As is described in the schematic of bacterial dynamics, during the early stage of infection, the rRNA is at a low transcriptional level where RNA-based tests show inferior performance in analytical sensitivity compared with DNA-based tests, while in active infection, the rRNA maintains a high transcriptional level so RNA-based NAATs can detect pathogens at much lower concentrations compared to DNA-based NAATs [21, 22, 37-39]. In the recovery phase, rRNA, as a marker of bacterial metabolic status, can represent replication and transcriptional activity, thus truly reflecting the infection situation [22, 23]."

Therefore, in terms of the analytical sensitivity of pathogen detection, RNA-based NAATs are more suitable for detection during active infection and recovery phase, rather than in the early stage of infection. Compared with RNA-based NAATs, DNA-based NAATs are more suitable for detection in the early stage of infection.” (Discussion, Page 15-16 line 318-337)

References:

- 21. Mouton, J.W., et al., Detection of Chlamydia trachomatis in male and female urine specimens by using the amplified Chlamydia trachomatis test. Journal of Clinical Microbiology, 1997. 35(6): p. 1369-1372.**
- 22. Burton, M.J., et al., Conjunctival chlamydial 16S ribosomal RNA expression in trachoma: is chlamydial metabolic activity required for disease to develop? Clinical infectious diseases, 2006. 42(4): p. 463-470.**
- 23. Engel, J.N. and D. Ganem, Chlamydial rRNA operons: gene organization and identification of putative tandem promoters. Journal of bacteriology, 1987. 169(12): p. 5678-5685.**
- 37. Mathews, S.A., K.M. Volp, and P. Timms, Development of a quantitative gene expression assay for Chlamydia trachomatis identified temporal expression of σ factors. FEBS letters, 1999. 458(3): p. 354-358.**
- 38. Peuchant, O., et al., Effects of antibiotics on Chlamydia trachomatis viability as determined by real-time quantitative PCR. Journal of medical microbiology, 2011. 60(4): p. 508-514.**
- 39. Wooters, M.A., et al., A real-time quantitative polymerase chain reaction assay for the detection of Chlamydia in the mouse genital tract model. Diagnostic microbiology and infectious disease, 2009. 63(2): p. 140-147.**

Comment 3: I also suggest the authors briefly highlight the main conclusion(s) of their results in the abstract and at the start of their discussion.

Answer 3: Thank you for your valuable suggestion.

As you pointed out, the conclusion wasn't fully highlighted in the abstract and discussion, so we have made emphasis on these sections. But out of consideration for overall framework and logical coherence in the discussion section, we have put the conclusion at the end of the discussion.

The modified content is as follows:

1. Abstract

“The LODs of the seven DNA kits for *CT*, *NG* and *UU* ranged between 38-1480, 94-20011, and 132-2011 copies/mL, respectively. As for RNA kits, they could detect samples at RNA concentrations of 3116, 2509 and 2896 copies/mL, respectively. The RNA concentrations of *CT*, *NG*, and *UU* were 40, 885, and 42 times that of corresponding pathogen DNA concentrations in the employed supernatants, so RNA kits could detect pathogen DNA concentrations as low as 78 copies/mL, 3 copies/mL, and 69 copies/mL, respectively, but the level of pathogen load that the RNA tests could detect was primarily dependent on the infectious phase and transcriptional level of RNA. Thus, a schematic of bacterial dynamics during the period of reproductive tract infections (RTIs) was provided, which suggesting that in terms of the analytical sensitivity of pathogen detection, RNA tests are more suitable for detecting active infection and recovery phase, while DNA tests are more suitable for detection in the early stage of infection.”

(Abstract, Page 2 line 29-41)

2. Discussion

“In conclusion, most DNA-based NAATs could detect *CT*, *NG*, and *UU* at DNA concentrations lower than 1000 copies/mL, while RNA-based NAATs could detect bacteria at RNA concentrations around 3000 copies/mL. When converting the RNA concentration level to pathogen DNA concentration level using the copy number ratios of RNA/DNA based on the employed supernatants, RNA detection kits exhibited satisfactory analytical sensitivity with LODs as low as 78 copies/mL, 3 copies/mL, and 69 copies/mL for *CT*, *NG* and *UU*, respectively, but the level of pathogen load that the RNA tests could detect was primarily dependent on the infectious phase and transcriptional level of RNA. As is

described in the schematic of bacterial dynamics, during the early stage of infection, the rRNA is at a low transcriptional level where RNA-based tests show inferior performance in analytical sensitivity compared with DNA-based tests, while in active infection, the rRNA maintains a high transcriptional level so RNA-based NAATs can detect pathogens at much lower concentrations compared to DNA-based NAATs [21, 22, 37-39]. In the recovery phase, rRNA, as a marker of bacterial metabolic status, can represent replication and transcriptional activity, thus truly reflecting the infection situation [22, 23]. Therefore, in terms of the analytical sensitivity of pathogen detection, RNA-based NAATs are more suitable for detection during active infection and recovery phase, rather than in the early stage of infection. Compared with RNA-based NAATs, DNA-based NAATs are more suitable for detection in the early stage of infection.” (Discussion, Page 15-16 line 318-337)

References:

21. Mouton, J.W., et al., Detection of Chlamydia trachomatis in male and female urine specimens by using the amplified Chlamydia trachomatis test. *Journal of Clinical Microbiology*, 1997. 35(6): p. 1369-1372.
22. Burton, M.J., et al., Conjunctival chlamydial 16S ribosomal RNA expression in trachoma: is chlamydial metabolic activity required for disease to develop? *Clinical infectious diseases*, 2006. 42(4): p. 463-470.
23. Engel, J.N. and D. Ganem, Chlamydial rRNA operons: gene organization and identification of putative tandem promoters. *Journal of bacteriology*, 1987. 169(12): p. 5678-5685.
37. Mathews, S.A., K.M. Volp, and P. Timms, Development of a quantitative gene expression assay for Chlamydia trachomatis identified temporal expression of σ factors. *FEBS letters*, 1999. 458(3): p. 354-358.
38. Peuchant, O., et al., Effects of antibiotics on Chlamydia trachomatis viability as determined by real-time quantitative PCR. *Journal of medical microbiology*, 2011. 60(4): p. 508-514.

39. Wooters, M.A., et al., A real-time quantitative polymerase chain reaction assay for the detection of Chlamydia in the mouse genital tract model. Diagnostic microbiology and infectious disease, 2009. 63(2): p. 140-147.

Comment 4: The "Importance" section of the abstract is almost exactly the same as the abstract, this needs to be modified according to the journal recommendations: "The Importance section should be no more than 150 words and should provide a nontechnical explanation of the significance of the study to the field. Avoid abbreviations and references and indicate the specific organism under study." (<https://journals.asm.org/journal/spectrum/article-types#research-articles>)

Answer 4: Thank you for your valuable suggestion.

According to your suggestion, we have modified the "Importance" section according to the journal recommendations. Nontechnical explanation of the significance of this study is provided; abbreviations and abbreviations and references are avoided, and the specific organism is indicated.

The modified content is as follows:

“Reproductive tract infections have considerable effects on the health of humans. *Chlamydia trachomatis*, *Neisseria gonorrhoeae*, and *Ureaplasma urealyticum* are common pathogens. Although evaluation of DNA-based tests has already existed, the comparison between DNA and RNA-based tests is rare. Therefore, this study compared limits of detection of the two tests on the same experimental conditions. Results suggested that most DNA-based NAATs could detect *Chlamydia trachomatis*, *Neisseria gonorrhoeae*, and *Ureaplasma urealyticum* at DNA concentrations lower than 1000 copies/mL, while RNA-based NAATs could detect bacteria at RNA concentrations around 3000 copies/mL. Considering the copy number of RNA per bacterium is dynamic through the growth cycle, further comparison is combined with a schematic of bacterial dynamics. Results suggested that in terms of the analytical sensitivity of pathogen detection, RNA tests are more suitable for detecting active infection and recovery phase, while

DNA tests are more suitable for detection in the early stage of infection.”

(Importance, Page 2-3 line 42-55)

Comment 5: Bacterial species names should be in italics throughout the text.

Answer 5: Thanks for pointing this out.

We are sorry for this mistake. According to your suggestion, all bacterial species names have been italicized and all amendments have been marked in red throughout the text.

Comment 6: In the abstract the RTI abbreviation is used before the use of the whole concept. This should be modified.

Answer 6: Thanks for pointing this out.

We are sorry for this mistake. According to your suggestion, we have added the definition of the term RTI in the abstract.

The modified content is as follows:

“Thus, a schematic of bacterial dynamics during the period of **reproductive tract infections (RTIs) was provided, which suggesting that in terms of the analytical sensitivity of pathogen detection, RNA tests are more suitable for detecting active infection and recovery phase, while DNA tests are more suitable for detection in the early stage of infection.”** (Abstract, Page 2 line 37-41)

Comment 7: Line 235: missing word: "the NG load of was 2.39..."

Answer 7: Thanks for pointing this out.

We are sorry for this mistake. According to your suggestion, we have revised the sentence. Besides, we found that the description of UU existed the same mistake, so we corrected them together.

The modified content is as follows:

“Considering there are four copies of 16S rDNA in the *NG* genome, **the pathogen load of *NG* was 2.39×10^8 copies/mL**, indicating that the concentration of 16S rRNA was around 885 times that of pathogen DNA in the employed supernatants. For *UU*, the original 16S rDNA and 16S rRNA concentrations were 5×10^6 and 1.05×10^8 copies/mL, respectively. Considering there are two copies of 16S rDNA gene in the *UU* genome, **the pathogen load of *UU* was 2.5×10^6 copies/mL**, indicating that the concentration of 16S rRNA was around 42 times that of pathogen DNA in the employed supernatants.” (Results, Page10 line 194-201)

Comment 8: Tables: All footnotes are missing the upper-case C in *Chlamydia trachomatis*

Answer 8: Thanks for pointing this out.

We are sorry for this mistake. According to your suggestion, we have capitalized the letter “C” in *Chlamydia trachomatis* in all footnotes of tables and all amendments have been marked in red throughout the text.

Comment 9: Table S2: The visual alignment between the Plasmid/Pathogen column and Primer/probe sequence column needs correction.

Answer 9: Thank you for your valuable suggestion.

To ensure neat visual alignment of the table, we have placed the content of the Plasmid/Pathogen column in the upper-left corner of each cell in the Supplementary Material Table S2.

The modified content is as follows:

Table S2 The primers and probes for DNA and RNA quantification

Plasmid/Pathogen	Target gene	Primer/Probe sequence (5' to 3')
Barcode plasmid	barcode sequence	QF: TTAATTACTGGGGGTCTTAGGC QR: CACCTACGAACCAAACCCAA

CT plasmid and CT	23S rDNA	QP: 6-FAM-ACGTCGATGGCTACATAAGAGG-BHQ1
		QF: TCCCTCGCCGTAAGCCCAAG
		QR: TTAAACCTGCTGCTCCATCGT
NG plasmid and NG	16S rDNA	QP: 6-FAM-CCAGGGTCAAGCTCGTCTTCC-BHQ1
		QF: CAGCTAATACCGCATAACGTCT
		QR: GCTACTGATCGTCGCCTTGGTG
UU plasmid and UU	16S rDNA	QP: 6-FAM-CCTTTACCCCGCCAACCAGCTA-BHQ1
		QF: CCCTAGTAGTCCACACCGTA
		QR: CCGTCAATTCCGTTTGAGTT
		QP: 6-FAM-ATGTGCCTGGGTAGTACATTCG-BHQ1

Abbreviations: *CT*, *Chlamydia trachomatis*; *NG*, *Neisseria gonorrhoeae*; *UU*, *Ureaplasma urealyticum*.

Comment 10: The text needs some minor English modifications as the use of "domestic", or the redaction of the last sentence of the paper: "As a matter of fact, both DNA and RNA-based NAATs exist inherent drawbacks such as missed detection...", which needs clarification, among others.

Answer 10: Thank you for your valuable suggestion.

We intended to reiterate the drawbacks of DNA and RNA-based NAATs that have been described in the discussion (Discussion, Page 15 line 298-317), and according to your suggestion, we have clarified and explained the description explicitly. Besides, we have polished some English expressions to make them more idiomatic after scrutinizing the whole article.

The modified content is as follows:

“As expatiated above, both DNA and RNA-based NAATs exist inherent drawbacks. Most DNA-based tests are targeted at plasmids, which may cause missed detection of plasmid-deficiency bacteria, and rRNA is subject to

mutations, thus contributing to the missed detection of variants. Therefore, the long-term concern of plasmid-deficiency strains for DNA-based tests and surveillance of variants for RNA-based tests are indispensable and essential.”

(Discussion, Page 16 line 337-343).

References:

1. Mitsunari, M., et al., *Cervical Ureaplasma urealyticum colonization might be associated with increased incidence of preterm delivery in pregnant women without prophlogistic microorganisms on routine examination*. Journal of Obstetrics and Gynaecology Research, 2005. **31**(1): p. 16-21.
2. Moridi, K., et al., *Epidemiology of genital infections caused by Mycoplasma hominis, M. genitalium and Ureaplasma urealyticum in Iran; a systematic review and meta-analysis study (2000–2019)*. BMC Public Health, 2020. **20**: p. 1-13.
3. Zahirnia, Z., et al., *Frequency of Chlamydia trachomatis, Mycoplasma genitalium, and Ureaplasma urealyticum isolated from vaginal samples of women in Kerman, Iran*. Archives of Clinical Infectious Diseases, 2018. **13**(6).
4. Huang, C., et al., *Mycoplasma and ureaplasma infection and male infertility: a systematic review and meta-analysis*. Andrology, 2015. **3**(5): p. 809-816.
5. Workowski, K.A., et al., *Sexually transmitted infections treatment guidelines, 2021*. MMWR Recommendations and Reports, 2021. **70**(4): p. 1.
6. Frenzer, C., et al., *Adverse pregnancy and perinatal outcomes associated with Mycoplasma genitalium: systematic review and meta-analysis*. Sexually transmitted infections, 2022. **98**(3): p. 222-227.

July 1, 2023

Dr. Rui Zhang

Graduate School, Peking Union Medical College, Chinese Academy of Medical Sciences, Beijing, People's Republic of China;
National Center for Clinical Laboratories, Beijing Hospital
Beijing
China

Re: Spectrum01497-23R1 (Comparison of analytical sensitivity of DNA-based and RNA-based nucleic acid amplification tests for reproductive tract infection pathogens: implications for clinical applications)

Dear Dr. Rui Zhang:

Your manuscript has been accepted, and I am forwarding it to the ASM Journals Department for publication. You will be notified when your proofs are ready to be viewed.

Sincerely,

Florence Doucet-Populaire
Editor, Microbiology Spectrum
